# FACTKB: Generalizable Factuality Evaluation using Language Models Enhanced with Factual Knowledge

**Shangbin Feng**[1]    **Vidhisha Balachandran**[2]    **Yuyang Bai**[3]    **Yulia Tsvetkov**[1]
[1]University of Washington    [2]Carnegie Mellon University    [3]Xi'an Jiaotong University
{shangbin, yuliats}@cs.washington.edu    vbalacha@cs.cmu.edu    1206944633@stu.xjtu.edu.cn

## Abstract

Evaluating the factual consistency of automatically generated summaries is essential for the progress and adoption of reliable summarization systems. Despite recent advances, existing factuality evaluation models are not robust, being especially prone to entity and relation errors in new domains. We propose FAC-TKB—a simple new approach to factuality evaluation that is generalizable across domains, in particular with respect to entities and relations. FACTKB is based on language models pretrained using facts extracted from external knowledge bases. We introduce three types of complementary factuality pretraining objectives based on entity-specific facts, facts extracted from auxiliary knowledge about entities, and facts constructed compositionally through knowledge base walks. The resulting factuality evaluation model achieves state-of-the-art performance on two in-domain news summarization benchmarks as well as on three out-of-domain scientific literature datasets. Further analysis of FACTKB shows improved ability to detect erroneous entities and relations in summaries and is robust and easily generalizable across domains. Code and data are available at https://github.com/BunsenFeng/FactKB.

## 1   Introduction

Generating factually accurate document summaries in addition to fluent and informative ones is critical to the adoption of summarization models (Kryś-ciński et al., 2020; Goyal and Durrett, 2020). However, evaluating the factual consistency of summaries is still challenging, especially in specialized domains like scientific or legal (Cachola et al., 2020; Goldsack et al., 2022; Polsley et al., 2016; Kanapala et al., 2019). The key reason is that the majority of existing approaches employ neural classifiers trained on synthetic data constructed from a relatively small set of documents (Kryściński et al., 2020; Goyal and Durrett, 2020). These factuality

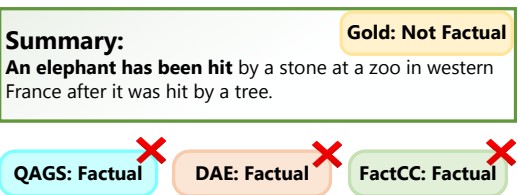

Figure 1: Existing factuality models struggle to identify *semantic frame errors* encompassing entities and relations. In the example, they fail to identify an error in the generated summary about *who* was hit by the stone.

classifiers are thus not robust to ever-growing information, in which the distribution of entities, events, and their relations changes greatly across time and domains (Elsahar and Gallé, 2019; Laparra et al., 2020). Pagnoni et al. (2021) highlighted this limitation, finding that over 50% of factuality errors in the XSUM (Narayan et al., 2018) summarization dataset stem from *semantic frame errors*, namely entities, events, and relations between them, as illustrated in Figure 1.

To address these issues, we develop a new factuality evaluation model with improved factual knowledge representation, specifically focusing on entities and relations. Entity-oriented pretraining objectives have been shown to improve QA and reasoning tasks (Yasunaga et al., 2022; Liu et al., 2022b); we thus hypothesize that similar objectives can aid factuality evaluation in better detecting semantic frame errors in generated summaries.

We propose FACTKB, a novel factuality evaluation model built upon language models (LMs) augmented with factual knowledge (§2). The LMs

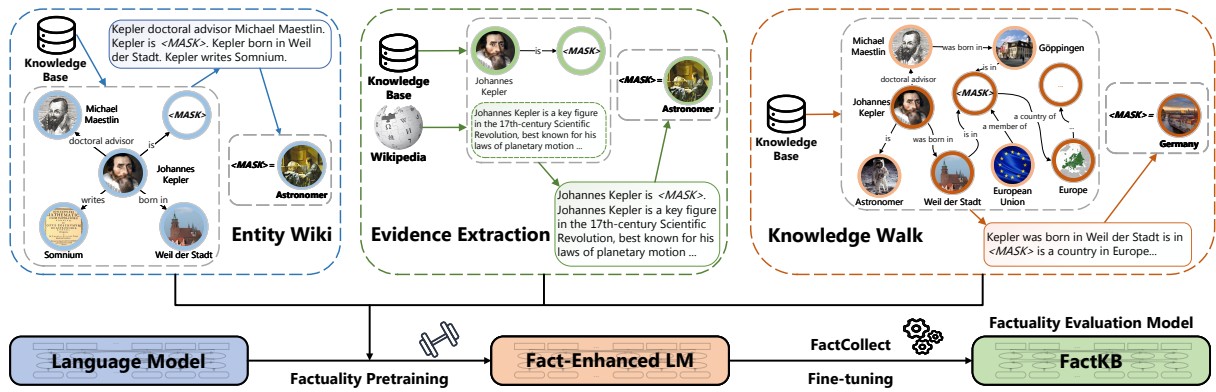

Figure 2: Overview of FACTKB. FACTKB pretrains LMs using three entity-centric pretraining strategies to improve fact representations. The objectives are designed to fill masked entities/relations in KB facts using i) *Entity Wiki* - direct facts about entities ii) *Evidence Extraction* - auxiliary knowledge about entities and iii) *Knowledge Walk* - compositional knowledge from the KB. The pretrained LMs are then fine-tuned for robust factuality evaluation.

are pretrained with knowledge-focused objectives using text synthesized from external knowledge bases (KBs) which store high-quality facts about entities and relations. We propose three types of complementary pretraining strategies: (1) **entity wiki**, with a focus on improving entity understanding; (2) **evidence extraction**, with a focus on incorporating supporting evidence from surrounding context; and (3) **knowledge walks**, with a focus on augmenting compositional reasoning about entities. For factuality evaluation, we first pretrain a language model using these three entity-centric pretraining strategies, and then fine-tune the enhanced LM on a factual error detection dataset.

We evaluate FACTKB's correlation with human factuality judgments across three settings (§3). In in-domain (news) summarization, FACTKB significantly outperforms baselines by 2–7 balanced accuracy (BACC) points on the FactCollect dataset (Ribeiro et al., 2022) and 10–12 correlation points on the FRANK benchmark (Pagnoni et al., 2021), particularly showing marked improvements in semantic frame errors. In out-of-domain experiments, FACTKB consistently outperforms existing approaches by 3–5 BACC points on three datasets in biomedical and scientific domains (Saakyan et al., 2021; Sarrouti et al., 2021; Wadden et al., 2020), demonstrating stronger generalizability to unseen documents in new domains. Further analysis shows that FACTKB is compatible with different LMs and KBs while presenting a lightweight and easy-to-use approach to factuality evaluation. Code, data, and trained factuality evaluation models are publicly available.

## 2 FACTKB Methodology

FACTKB aims to improve the robustness and generalizability of factuality evaluation by a simple *factuality pretraining*, which improves entity and relation representations in LMs. We first propose three pretraining strategies (§2.1). We then describe the training process to (1) pretrain an LM using the proposed strategies and (2) fine-tune the fact-enhanced LM on a factuality error detection dataset, resulting in FACTKB (§2.2). Figure 2 presents an overview of our approach.

### 2.1 Factuality Pretraining

Knowledge bases are rich reservoirs of facts about entities and relations (Vrandečić and Krötzsch, 2014; Pellissier Tanon et al., 2020), and we explore the possibility of leveraging external KBs as "fact teachers" to enhance an LM's representation of entities and relations.

Let $\text{KB} = (\mathcal{E}, \mathcal{R}, \mathbf{A}, \epsilon, \varphi)$, where $\mathcal{E} = \{e_1, \ldots, e_\mathcal{N}\}$ represents the entities in the KB, $\mathcal{R} = \{r_1, \ldots, r_\mathcal{M}\}$ denotes the relations in the KB, $\mathbf{A}$ denotes the adjacency matrix where $a_{ij} = k$ indicates relation $r_k$ connecting entities $e_i$ and $e_j$ $(e_i, r_k, e_j) \in \text{KB}$, $\epsilon(\cdot) : \mathcal{E} \to \text{str}$ and $\varphi(\cdot) : \mathcal{R} \to \text{str}$ map the entities and relations to their textual names. We propose three novel types of factuality pretraining strategies that leverage the KB.

**Strategy 1: Entity Wiki** Entities in KBs often have multiple edges connecting them to other entities via relations, each representing a distinct but related fact about the entity. Inspired by the task of knowledge base completion (Bordes et al., 2013; Vashishth et al., 2019) to predict missing connec-

| Factuality Pretraining | Corpus Size Bound | # Tokens | Example |
|---|---|---|---|
| ENTITY WIKI | $\propto |\mathcal{E}|$ | 5.4M | Johannes Kepler is born in Italy. Johannes Kepler is an [MASK]. [SEP] Johannes Kepler is the author of Astronomia nova. . . . |
| EVIDENCE EXTRACTION | $\propto ||A||_0$ | 12.2M | Hillary Clinton party affiliation [MASK] Hillary Diane Rodham Clinton is an American politician, . . . Member of the Democratic Party, she was the nominee . . . |
| KNOWLEDGE WALK | $\propto |\mathcal{E}|(\frac{||A||_0}{|\mathcal{E}|})^k$ | 2.7M | University of Edinburgh located in Scotland located in [MASK] is a continent . . . |

Table 1: Summary of the three factuality pretraining strategies.

tions in KBs based on available KB facts, we propose the *entity wiki* factuality pretraining, where an LM is pretrained with the task of predicting masked entities or relations in KB facts. Specifically, for each entity $e_i \in \mathcal{E}$, we retrieve its one-hop neighborhood in the KB as $\mathcal{E}_{e_i} = \{e_j \mid \exists r_k \ s.t. \ a_{ij} = k\}$. We then synthesize a sentence using entity $e_i$ and its connected one-hop facts:

$$\boldsymbol{d}_i = \text{concat}_{e_j \in \mathcal{E}_{e_i}} \big[ \epsilon(e_i)\varphi(r_k|a_{ij}=k)\epsilon(e_j)[\text{SEP}] \big]$$

where concat denotes string concatenation and $[SEP]$ denotes the special token. Repeating this generation process for all $e \in \mathcal{E}$, we produce a corpus of entity facts as $\{\boldsymbol{d}_i\}_{i=1}^{|\mathcal{E}|}$ with the max size being the number of entities $|\mathcal{E}|$. We use this entity wiki corpus to pretrain an LM for better factual reasoning by randomly masking entities and relations in it and training the LM to predict the mask given the surrounding facts about an entity. We randomly mask the corpora with probability $p$ and pretrain LMs with the masked language modeling objective. We expect this objective to train LMs to infer facts from surrounding knowledge and penalize unsupported hallucinations about entities and relations.

**Strategy 2: Evidence Extraction** The goal of this pretraining strategy is to enhance the model's ability to evaluate facts based on relevant evidence. We begin by randomly selecting a triple $(e_i, r_k, e_j) \in \text{KB}$ and use the first paragraph of the Wikipedia description of $e_i$ as the auxiliary knowledge. We synthesize a sentence using the two as:

$$\boldsymbol{d}_i = \epsilon(e_i) \ \varphi(r_k) \ [\text{MASK}] \ \text{Wikipedia}(e_i)$$

where we mask out $\epsilon(e_j)$ and [MASK] denotes the special token, Wikipedia$(\cdot) : \mathcal{E} \to \text{str}$ maps entities to the first paragraph of their Wikipedia description. Repeating this process $N$ times with randomly selected triples, we obtain a corpus of triples paired with auxiliary knowledge $\{\boldsymbol{d}_i\}_{i=1}^{N}$. The corpus size is bounded by all KB triples represented as

the $L_0$ norm of the adjacency matrix $||A||_0$. We use this corpus for the evidence extraction factuality pretraining and train the LM to predict the mask by using relevant evidence in the auxiliary paragraph. Through this, we aim to augment FACTKB's ability to implicitly select evidence from the document to support its factuality evaluation.

**Strategy 3: Knowledge Walk** Natural language documents often include compositional statements about entities and relations (Feldman and El-Yaniv, 2019; Wang and Pan, 2022), but pretrained LMs struggle with such compositional reasoning (Press et al., 2022). To improve FACTKB's ability to understand multi-hop claims, we propose the *knowledge walk* factuality pretraining strategy. Specifically, we randomly select a starting entity $e_{(0)}$ and randomly select an entity $e_{(1)}$ from its direct neighborhood $\mathcal{E}_{e_{(0)}}$, resulting in a one-hop triple $\{e_{(0)}, r_{(0,1)}, e_{(1)}\}$ where $r_{(0,1)}$ denotes the relation between $e_{(0)}$ and $e_{(1)}$. Now, from $e_{(1)}$, we randomly select an entity from it's direct neighborhood to take the next step. We repeat this process for $\mathcal{K}$ times, and obtain a $\mathcal{K}$-hop random walk of triples beginning at $e_{(0)}$: $\{e_{(0)}, r_{(0,1)}, e_{(1)}, \cdots, r_{(\mathcal{K}-1, \mathcal{K})}, e_{(\mathcal{K})}\}$. We then produce a sentence based on the $\mathcal{K}$-hop walk:

$$\boldsymbol{d}_i = \epsilon(e_{(0)}) \, \text{concat}_{i=0}^{\mathcal{K}-1} \big[ \, \varphi(r_{(i,i+1)}) \, \epsilon(e_{(i+1)}) \, \big]$$

Repeating this $\mathcal{K}$-hop walk $N$ times with different randomly selected starting entities, we obtain $\{\boldsymbol{d}_i\}_{i=1}^{N}$ as the corpus for the knowledge walk factuality pretraining, whose size is bounded by the number of all possible $\mathcal{K}$-hop walks as $|\mathcal{E}|(\frac{||A||_0}{|\mathcal{E}|})^k$. In this corpus, we randomly mask entities or relations in each group of facts with probability $p$ and train an LM to predict the masked element using the compositional facts around it using the masked language model objective. Through this pretraining, we expect FACTKB to improve in compositional

| Model | All Data | | CNN/DM | | XSUM | |
|---|---|---|---|---|---|---|
| | **BACC** | **F1** | **BACC** | **F1** | **BACC** | **F1** |
| QAGS | 79.8 | 79.7 | 64.2 | 76.2 | 59.3 | 85.2 |
| QUALS | 78.3 | 78.5 | 60.8 | 76.2 | 57.5 | 82.2 |
| ROBERTA | 76.1 | 76.5 | 62.5 | 76.2 | 62.1 | 78.3 |
| FALSESUM | 78.9 | 78.2 | 53.7 | 34.6 | 61.1 | 64.3 |
| FALSESUM+ | 84.2 | 83.7 | 64.2 | 77.1 | 67.4 | 82.1 |
| SUMMAC | 86.6 | 86.2 | 75.4 | 83.5 | 71.9 | 90.4 |
| FACTCC | 76.0 | 76.3 | 69.0 | 77.8 | 55.9 | 73.9 |
| FACTCC+ | 83.9 (±0.4) | 84.2 (±0.4) | 68.0 (±1.0) | 83.7 (±0.5) | 58.3 (±2.2) | 84.0 (±1.0) |
| FACTGRAPH | 86.3 (±1.3) | 86.7 (±1.1) | 73.0 (±2.3) | 86.8 (±0.8) | 68.6 (±2.3) | 86.6 (±2.0) |
| FACTGRAPH-ADAPTERS | 87.6 (±0.7) | 87.8 (±0.7) | 76.0 (±2.8) | 87.5 (±0.4) | 69.9 (±2.3) | 88.4 (±1.2) |
| FACTKB-WIKI | 89.3 (±0.4)* | **89.5** (±0.5)* | 77.3 (±0.3)* | **88.2** (±0.6)* | 77.3 (±1.3)* | **91.8** (±1.2)* |
| FACTKB-EVIDENCE | **89.4** (±0.2)* | **89.5** (±0.3)* | 77.7 (±1.4)* | 87.9 (±0.7) | 76.8 (±1.9)* | 90.8 (±0.8)* |
| FACTKB-WALK | 89.1 (±0.4)* | 89.3 (±0.5)* | **78.3** (±1.2)* | 87.7 (±0.4) | 76.4 (±0.3)* | 90.4 (±1.4)* |

Table 2: Performance of FACTKB on the FactCollect dataset. We report average performance and standard deviation across 5 random seeds. Best performance is shown in **bold**, while * indicates statistical significance. FACTKB significantly outperforms existing factuality evaluation approaches on in-domain evaluation.

fact understanding about entities and relations appearing in the summary and the input document.

We briefly summarize the three factuality pretraining strategies and provide examples in Table 1.

## 2.2 FACTKB Training

We initialize FACTKB with encoder-based LMs and pretrain FACTKB separately with each of the three factuality pretraining corpora using the masked language modeling objective to study the effectiveness of each strategy. This results in fact-enhanced LMs with the ability to better represent facts, entities, and relations. Finally, we fine-tune FACTKB on human-annotated factual error detection datasets with the sequence classification setting, taking SUMMARY [SEP] DOCUMENT as input and produce FACTUAL or NON-FACTUAL labels. The [CLS] token is adopted for classification. As a result, we obtain FACTKB, our entailment-based factuality evaluation model that classifies machine-generated summaries as factual or non-factual.

## 3 Data and Experiment Settings

### 3.1 Training

**Data** We use YAGO (Pellissier Tanon et al., 2020), an encyclopedic knowledge base based on Wikidata (Vrandečić and Krötzsch, 2014), to construct the three types of factuality pretraining corpora, while we discuss FACTKB's compatibility with different KBs in Section 5.2. For finetuning, we use the FactCollect dataset (Ribeiro et al., 2022), a dataset for factual error detection that gathers human annotations from different sources (Wang et al., 2020a; Kryściński et al., 2020; Maynez et al.,

2020; Pagnoni et al., 2021) and consolidates them into a single dataset. It mainly focuses on the news media domain, covering summaries and articles from CNN, Daily Mail, and BBC. FactCollect follows a binary classification setting where each (SUMMARY, ARTICLE) pair has a FACTUAL or NON-FACTUAL label. We present more details about the FactCollect dataset in Appendix C.

**Settings** We use a ROBERTA-BASE (Liu et al., 2019) checkpoint and continue pretraining separately on each of the three factuality pretraining corpora. We discuss FACTKB's compatibility with different LM initializations in Section §5.2. We assign corpus size parameter $N = 1e5$, masking probability $p = 0.15$, and knowledge walk length $\mathcal{K} = 5$ in the experiments, while we discuss the effect of corpus size and knowledge walk length in Appendix 5.4. We use a learning rate of $2e-5$ for pretraining, $1e-4$ for fine-tuning, a batch size of 32, and the RAdam optimizer. Pretraining is conducted for 5 epochs and fine-tuning has 50 maximum epochs with early stopping. More hyperparameter settings are presented in Appendix 3.1.

**Hyperparameters** We propose to further pretrain LM checkpoints with three types of factuality pretraining and fine-tune on factuality evaluation datasets. We present hyperparameters for the pretraining and fine-tuning stage in Table 4. We mostly follow the hyperparameters in Gururangan et al. (2020) for the pretraining stage. The default hyperparameters on Huggingface Transformers are adopted if not included in Table 4.

| Model | All Data | | | | CNN/DM | | | | XSUM | | | |
|---|---|---|---|---|---|---|---|---|---|---|---|---|
| | $\rho$ | p-val | r | p-val | $\rho$ | p-val | r | p-val | $\rho$ | p-val | r | p-val |
| QAGS | .22 | .00 | .23 | .00 | .34 | .00 | .27 | .00 | .07 | .05 | .06 | .09 |
| QUALS | .22 | .00 | .19 | .00 | .31 | .00 | .27 | .00 | .14 | .00 | .07 | .03 |
| DAE | .17 | .00 | .20 | .00 | .27 | .00 | .22 | .00 | .03 | .38 | .33 | .00 |
| ROBERTA | .35 | .00 | .41 | .00 | .43 | .00 | .31 | .00 | .23 | .00 | .15 | .00 |
| FALSESUM | .05 | .00 | .04 | .11 | .07 | .05 | .07 | .03 | .04 | .28 | .04 | .35 |
| FALSESUM+ | .22 | .00 | .26 | .00 | .27 | .00 | .33 | .00 | .24 | .00 | .27 | .00 |
| SUMMAC | .33 | .00 | .35 | .00 | .42 | .00 | .36 | .00 | .24 | .00 | .25 | .00 |
| FACTCC | .20 | .00 | .29 | .00 | .36 | .00 | .30 | .00 | .06 | .07 | .19 | .00 |
| FACTCC+ | .32 | .00 | .38 | .00 | .40 | .00 | .28 | .00 | .24 | .00 | .16 | .00 |
| FACTGRAPH | .35 | .00 | .42 | .00 | .45 | .00 | .34 | .00 | .30 | .00 | **.49** | .00 |
| FACTKB-WIKI | .46 | .00 | **.52** | .00 | **.57** | .00 | **.49** | .00 | .29 | .00 | .39 | .00 |
| FACTKB-EVIDENCE | .43 | .00 | .49 | .00 | .53 | .00 | .45 | .00 | .31 | .00 | .37 | .00 |
| FACTKB-WALK | **.47** | .00 | **.52** | .00 | **.57** | .00 | .45 | .00 | **.35** | .00 | .36 | .00 |

Table 3: Correlation of FACTKB with human judgments of factuality on the FRANK benchmark. Best performance is shown in **bold**. FACTKB has the highest correlation with human judgments across five of the six settings.

| Pretraining Stage | | Fine-Tuning Stage | |
|---|---|---|---|
| Hyperparameter | Value | Hyperparameter | Value |
| LEARNING RATE | $2e$-5 | LEARNING RATE | $1e$-4 |
| WEIGHT DECAY | $1e$-5 | WEIGHT DECAY | $1e$-5 |
| MAX EPOCHS | 5 | MAX EPOCHS | 50 |
| BATCH SIZE | 32 | BATCH SIZE | 32 |
| OPTIMIZER | ADAM | OPTIMIZER | RADAM |
| ADAM EPSILON | $1e$-6 | | |
| ADAM BETA | 0.9, 0.98 | | |
| WARMUP RATIO | 0.06 | | |
| EVIDENCE: $N$ | $1e5$ | | |
| WALK: $N$ | $1e5$ | | |
| WALK: $\mathcal{K}$ | 5 | | |

Table 4: Hyperparameter settings of FACTKB.

## 3.2 Evaluation

To study the robustness of FACTKB, we perform both in-domain and out-of-domain evaluation.

**In-Domain Evaluation** Since most research and resources on summarization and factuality are in the news media domain, we leverage the FactCollect dataset (Ribeiro et al., 2022) and the FRANK benchmark (Pagnoni et al., 2021) for in-domain factuality evaluation. We evaluate FACTKB on the held-out test set of the FactCollect dataset. FRANK (Pagnoni et al., 2021) is a factuality evaluation benchmark with human judgments on the factual consistency of model-generated summaries collected across 9 summarization models along with human annotations on the category of factual errors. Following the FRANK benchmark guidelines, we use two correlation measures (Pearson (Benesty et al., 2009) and Spearman (Myers and Sirois, 2004)). We present more details about the FRANK benchmark in Appendix C. Following previous work (Ribeiro et al., 2022), we train FACTKB

on the FactCollect dataset without the FRANK subset for the FRANK evaluation.

**Generalizable Factuality Evaluation** Summarization systems are used in diverse domains in the real world, including but not limited to news media (Liu et al., 2022c; Eyal et al., 2019; Li et al., 2016), social media (Syed et al., 2019; Kano et al., 2018; He et al., 2020), and scientific literature (Cachola et al., 2020; Lev et al., 2019). Consequently, factuality metrics should also provide reliable factuality scores in the face of shifting domains. To study this, we perform an out-of-domain evaluation using unseen documents and summaries from the scientific domain. To establish a test bed for generalizable factuality evaluation, we make use of three datasets in the scientific literature domain:

- **CovidFact** (Saakyan et al., 2021) collects claims from the r/COVID19 subreddit and verifies them against relevant scientific literature and Google search results, resulting in a binary classification setting that is similar to the FactCollect dataset.

- **HealthVer** (Sarrouti et al., 2021) consists of claims sourced from TREC-COVID (Voorhees et al., 2021) and verified against the CORD-19 (Wang et al., 2020b) corpus. While HealthVer originally follows a three-way classification setting (SUPPORT, REFUTE, NOT ENOUGH INFORMATION), we remove the examples in the "NOT ENOUGH INFORMATION" category to evaluate models as they are trained on the binary classification setting (factual, non-factual).

| Model | CovidFact | | HealthVer | | SciFact | |
|---|---|---|---|---|---|---|
| | **BACC** | **F1** | **BACC** | **F1** | **BACC** | **F1** |
| RANDOM | 52.7 | 41.3 | 46.8 | 53.0 | 49.0 | 57.5 |
| FACTCC | 52.3 | 49.2 | 51.8 | 51.9 | 42.7 | 45.9 |
| FACTCC+ | 51.1 | 50.5 | 49.5 | 51.6 | 48.6 | 55.2 |
| FACTGRAPH | 57.6 | 53.5 | 55.1 | 24.3 | 61.0 | 42.2 |
| FACTGRAPH-EDGE | 50.6 | 48.4 | 50.6 | 53.5 | 56.7 | 68.2 |
| FALSESUM | 50.6 | 41.6 | 56.8 | 51.2 | 45.7 | 65.3 |
| FALSESUM+ | 50.1 | 41.2 | 57.3 | 51.6 | 51.9 | 65.4 |
| SUMMAC | 57.6 | 53.4 | 52.5 | 41.2 | 59.8 | 46.9 |
| ROBERTA | 59.0 (±3.2) | 46.4 (±4.3) | 55.0 (±2.2) | 50.0 (±3.9) | 58.1 (±4.0) | 71.3 (±3.5) |
| FACTKB-WIKI | **64.8** (±0.3)* | **54.4** (±0.7)* | **60.1** (±0.4)* | **71.6** (±2.9)* | 62.9 (±0.4)* | 72.3 (±1.1)* |
| FACTKB-EVIDENCE | 63.9 (±0.6)* | 53.3 (±1.7) | 59.0 (±1.0)* | 70.8 (±0.9)* | 61.4 (±0.5)* | **74.1** (±1.6)* |
| FACTKB-WALK | 63.7 (±1.0)* | 53.1 (±1.6) | 58.5 (±0.5)* | 68.7 (±1.7)* | **63.1** (±1.1)* | 67.6 (±4.1) |

Table 5: Performance of FACTKB on out-of-domain scientific datasets. We report average performance and standard deviation across 5 random seeds. Best performance is shown in **bold**, while * indicates statistical significance. FACTKB exhibits better generalization to new domains across all three datasets.

- **SciFact** (Wadden et al., 2020) includes claims sourced from citation sentences in biomedical literature and verified against the cited paper's abstract. While SciFact uses three-way classification that includes "NOT ENOUGH INFORMATION", we similarly remove them in this work.

We leverage the well-organized version of the three datasets in Wadden et al. (2022). [1] We train and validate FACTKB with the FactCollect dataset from the news domain and evaluate on the test set of these datasets for zero-shot transfer learning.

**Baselines** We compare FACTKB with different types of existing factuality evaluation models: QAGS (Wang et al., 2020a), QUALS (Nan et al., 2021), DAE (Goyal and Durrett, 2020), FalseSum (Utama et al., 2022), SummaC (Laban et al., 2022), FactCC (Kryściński et al., 2020), and FactGraph (Ribeiro et al., 2022). Since training data is a key factor in factuality evaluation models and they are often used off-the-shelf, we include factuality evaluation measures trained on both synthetic data (QAGS, QUALS, DAE, SummaC, FalseSum, FactCC) and human-annotated data (RoBERTa, FalseSum+, FactCC+, FactGraph, FactGraph-edge). We follow the same train/dev/test dataset split and experiment settings so that the results are directly comparable. We present more details about the baselines in Appendix D.

## 4 Results

**In-Domain Results** We evaluate FACTKB and baselines on the FactCollect dataset using the en-

tire held-out test data, the CNN/DM subset and the XSUM (BBC) subset, and report balanced accuracy scores and micro F1 scores. We run each method five times with different random seeds and report the average performance as well as the standard deviation. Table 2 demonstrates that FACTKB significantly (*) outperforms all baseline factuality evaluation methods by 3.8 BACC points on average across the three dataset settings. This demonstrates that the introduction of KBs and factuality pretraining is beneficial for factuality evaluation. Among the three factuality pretraining strategies, all of them outperform baseline models, suggesting that FACTKB's general methodology is compatible with different types of KB utilization.

**Human Correlation** We evaluate FACTKB and baselines on the FRANK benchmark to study how well FACTKB correlates with human judgments. We use the official script [2] to report the Pearson ($\rho$) and Spearman ($r$) correlation and p-values. Results in Table 3 show that classification-based metrics (FactCC, FactGraph, and FACTKB) generally outperform QA-based metrics (QAGS and QUALS). FACTKB significantly advances the state-of-the-art on the FRANK benchmark, resulting in the improvement of 5-15 correlation points across multiple settings. Our results show that FACTKB is highly correlated with human judgments, making it a practical approach for evaluating the factual consistency of generated news summaries.

**Out-of-Domain Results** We evaluate FACTKB and existing factuality evaluation models on out-of-

---

[1] Dataset statistics are presented in Table 8.

[2] https://github.com/artidoro/frank

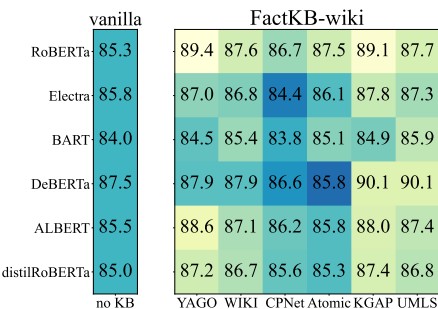

Figure 3: Compatibility of FACTKB across various LMs and KBs. We report BACC scores of different setups on the FactCollect dataset. FACTKB is a general method compatible with various LM and KB settings.

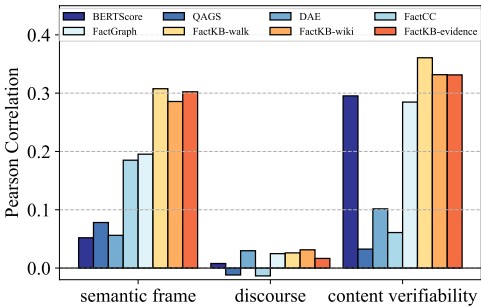

Figure 4: Correlation of FACTKB and baselines with human judgments across error categories. FACTKB shows significant improvement in capturing semantic frame errors and has slightly better or on-par performance on discourse and content verifiability errors.

domain scientific literature datasets in a zero-shot manner. Results are presented in Table 5, which demonstrate that while existing factuality evaluation models previously achieve good performance in the in-domain setting, they exhibit severe performance drops on the three out-of-domain datasets, performing only slightly better than random factuality scores (RANDOM). This suggests that existing approaches are not generalizable to other domains, limiting their applicability. On the contrary, FACTKB significantly (*) outperforms existing factuality metrics by 4.1 BACC points on average across the three out-of-domain datasets. Our results suggest that the factuality pretraining strategies enable FACTKB to better represent facts (entities and relations) in a new domain, making the factuality evaluation model more robust to shifting domains.

## 5 Analysis and Discussion

### 5.1 Where did FACTKB Improve?

To better understand FACTKB's improvement over existing approaches, we leverage the factual error typology in the FRANK benchmark (Pagnoni et al.,

2021) and examine FACTKB's performance on the three error categories: semantic frame, discourse, and content verifiability errors. Using the official script in the FRANK benchmark, we remove each category of errors and report changes in correlation scores. Higher variation indicates a greater influence on a model's ability to handle a certain type of error. Figure 4 demonstrates that FACTKB is significantly better at identifying semantic frame errors, which focus on entities and relations. This indicates that our KB-based factuality pretraining strategies successfully result in a better understanding of the facts regarding entities and relations. FACTKB also has good performance in other categories, resulting in a factuality evaluation model that captures diverse types of errors and advances the state-of-the-art across the board. We conduct further qualitative analysis in Appendix B.

### 5.2 KB and LM Compatibility

FACTKB uses pretrained LMs for initialization, leverages external KBs for factuality pretraining, and trains on factuality evaluation datasets to result in a factuality metric. Our general methodology to leverage knowledge bases as fact teachers for generalizable factuality evaluation could work with different LM and KB combinations. To study whether out approach works across different settings, we apply the FACTKB methodology to six different LMs (RoBERTa, Electra, BART, DeBERTa, ALBERT, and distilRoBERTa) and pretrain the LMs on factuality pretraining corpora constructed based on six different KBs (YAGO, Wikidata, ConceptNet, Atomic, KGAP, and UMLS). For each combination, we initialize a particular LM and pretrain it using the proposed three pretraining strategies based on a particular KB. For each setting, we evaluate the resulting model using the FactCollect dataset and report the BACC scores. We present the performance

| Metric | Pearson | Spearman | Usage Steps |
|---|---|---|---|
| QAGS | .22 | .23 | 1) extract answer candidates 2) generate the questions 3) answer the generated questions 4) compare the answers to obtain the QAGS factuality score |
| QUALS | .22 | .19 | 1) generating question and answer pairs from summaries 2) filter the generated question and answer for high-quality pairs 3) evaluate the generated question and answer pairs using the source document as input, compute QUALS scores for each summary |
| DAE | .17 | .20 | 1) preprocess summaries and documents with dependency parsing 2) run the pretrained model to get DAE scores |
| FACTCC | .20 | .29 | 1) run the pretrained model to get FactCC scores |
| FACTGRAPH | .35 | .42 | 1) build abstract meaning representation graphs 2) run the pretrained model to get FactGraph scores |
| FACTKB | **.47** | **.52** | 1) run the pretrained model to get FACTKB scores |

Table 6: Usage steps of factuality metrics and their performance on the FRANK benchmark. FACTKB (WALK) presents a state-of-the-art factuality metric with minimum hassle when evaluating new summaries and articles.

of different settings in Figure 3, which illustrates that regardless of which LM and KB, FACTKB generally results in improved factual error detection capabilities compared to the vanilla LM checkpoints without factuality pretraining. In addition, certain LMs (RoBERTa and DeBERTa) and KBs (YAGO, KGAP, and UMLS) are better than others, suggesting that the choice of the base LM and external KB warrants further research. Our results demonstrate that FACTKB is a general pretraining approach that can be applied to various LM-KB combinations to improve fact representations and develop better factuality evaluation models.

## 5.3 Simplicity Study

While existing factuality evaluation approaches require additional processing (such as computing the dependency structure (Goyal and Durrett, 2020) and AMR graphs (Ribeiro et al., 2022) or running multiple iterations of question generation (Fabbri et al., 2022)) in the face of new data, FACTKB requires no preprocessing and only uses a fine-tuned RoBERTa for sequence classification. We summarize the steps involved in using existing approaches and their performance on the FRANK benchmark in Table 6, which demonstrates that FACTKB not only has state-of-the-art performance but is also a lightweight and simple factuality evaluation model.

## 5.4 Parameter Analysis

**Corpus size.** For evidence extraction and knowledge walk, the pretraining corpus size $N$ is controllable and governs the amount of information towards augmenting FACTKB's ability towards factual errors regarding entities and relations. While we adopted $N = 1e5$ in the main experiments,

we further explore the effect of factuality pretraining corpus size in Figure 5. It is illustrated that $N = 1e4$ or $N = 1e5$ are generally desirable settings, while factuality pretraining with too large $N$s might be counterproductive. This could in part be attributed to catastrophic forgetting (Ramasesh et al., 2021), which warrants further research.

**Pretraining epoch.** FACTKB further pretrains LM checkpoints on the three factuality pretraining corpora, while the training epoch governs the intensity of such exercises. We adopted 5 epochs of continued pretraining in the main experiments, while we further explore the effect of pretraining epochs in Figure 5. it is demonstrated that 1 to 10 epochs are generally desirable while exercising too much might be counterproductive.

**Knowledge walk length.** An important aspect of the knowledge walk factuality pretraining is the generated walk length $\mathcal{K}$, which governs the degree of compositionality in the pretraining corpus. While we adopted $\mathcal{K} = 5$ in the main experiments, we further explore the effect of $\mathcal{K}$ in Figure 5. It is illustrated that $\mathcal{K} = 5$ performs best by providing a moderate amount of compositionality in the factuality pretraining corpora.

## 6 Related Work

**Factuality Evaluation** Recent advances in text summarization have presented models and systems that are capable of generating increasingly fluent, controllable, and informative summaries of documents (Liu and Lapata, 2019; Balachandran et al., 2021; Meng et al., 2022; Tang et al., 2022; Goldsack et al., 2022; Peng et al., 2021; Aharoni et al.,

2023; Liu et al., 2022d; Rothe et al., 2021; Narayan et al., 2021; Bhattacharjee et al., 2023; Chen et al., 2023b; He et al., 2023; Liu et al., 2023b; Chen et al., 2023a). However, they suffer from hallucination and might not be factually faithful towards the source document (Cao et al., 2018; Pagnoni et al., 2021; Balachandran et al., 2022; Tang et al., 2023; Liu et al., 2023a; Luo et al., 2023), leading to increased research in factuality evaluation. QA-based approaches (Wang et al., 2020a; Nan et al., 2021; Scialom et al., 2021; Fabbri et al., 2022) attempt to generate and answer questions based on summaries and documents and judge the factuality by comparing answers. Later approaches are generally entailment-based (Kryściński et al., 2020; Goyal and Durrett, 2020, 2021; Laban et al., 2022; Ribeiro et al., 2022), proposing to classify (summary, document) pairs into FACTUAL or NON-FACTUAL labels. Among them, FactCC (Kryściński et al., 2020) is one of the first entailment-based metrics and is trained on synthetic data; DAE (Goyal and Durrett, 2020, 2021) proposes to leverage the dependency structure of summaries and documents; FactGraph (Ribeiro et al., 2022) builds abstract meaning representation graphs and adopts graph neural networks for joint representation learning along the textual content. In addition, hypothesis re-ranking (Garneau and Lamontagne, 2021), counterfactual estimation (Xie et al., 2021), NLI models (Utama et al., 2022), phrase-level localization (Takatsuka et al., 2022), and weighting facts in the source document (Xu et al., 2020) were also explored in factuality evaluation. Moving beyond a binary concept of factuality, FRANK (Pagnoni et al., 2021) promotes a fine-grained understanding of factuality and proposes a typology of factuality errors. Inspired by its analysis that *semantic frame errors*, errors regarding entities and relations, are a major source of factuality errors yet under-explored by existing factuality metrics, we propose FACTKB to leverage external KBs for factuality pretraining and help enforce better factuality towards entities and relations discussed in summaries and documents.

**Knowledge Bases in NLP**    Knowledge base is a standard format for structured knowledge representation. One application of KBs in NLP is to inject knowledge and augment LMs, where different approaches focused aspects such as pretraining (Chen et al., 2020; Agarwal et al., 2021; Rosset et al., 2020; Li et al., 2022), document graphs (Hu et al., 2021; Zhang et al., 2022a), KB structure (Yasunaga

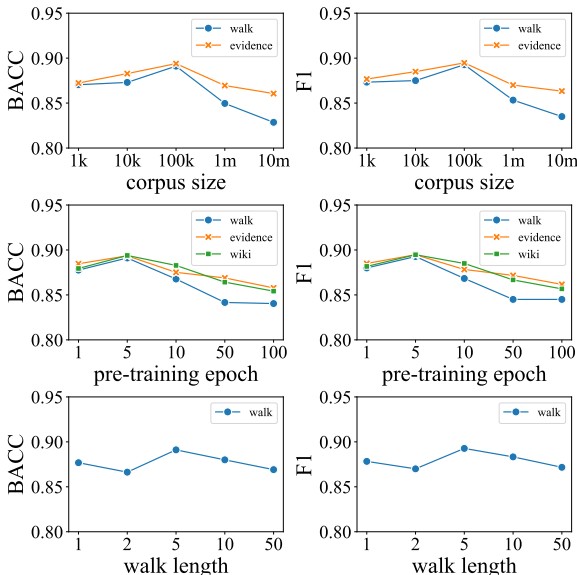

Figure 5: Parameter analysis of pretraining corpus size, epoch, and knowledge walk length.

et al., 2021; Zhang et al., 2022b), and long documents (Feng et al., 2023). KB-enhanced approaches also advanced numerous NLP tasks, ranging from question answering (Mitra et al., 2022; Bosselut et al., 2021; Oguz et al., 2022; Feng et al., 2022; Heo et al., 2022; Ma et al., 2022), text generation (Rony et al., 2022; Dognin et al., 2021; Yu et al., 2021), and commonsense reasoning (Kim et al., 2022; Jung et al., 2022; Amayuelas et al., 2021; Liu et al., 2022a). In this work, we tap into KBs' nature as high-quality reservoirs of factual information and construct factuality pretraining objectives to augment factuality evaluation.

## 7    Conclusion

We propose FACTKB, a simple and novel approach to factuality evaluation using language models pretrained on facts from external KBs to improve entity and relation representations. Specifically, we leverage KBs to construct three factuality pretraining objectives: entity wiki, evidence extraction, and knowledge walk. FACTKB pretrains an LM using the three objectives and fine-tunes the resulting model on factuality evaluation datasets. Extensive experiments demonstrate that FACTKB advances the state-of-the-art in both in-domain and out-of-domain factuality evaluation, better correlates with human factuality annotations, and better detects semantic frame errors. FACTKB presents an easy-to-use and generalizable factuality metric, facilitating research on factually-consistent summarization.

## Limitations

**LM and KB Selection**   While Section 5.2 offers empirical evidence that FACTKB is compatible with 6 language models and 6 external knowledge bases, it remains unclear upfront which LM and KB combination would be most desirable. While empirical performance could be a good guide, there are several unaddressed possibilities: For language models, it is possible to leverage an ensemble of FACTKBs seeded with different LM checkpoints and architectures. This might result in better factuality evaluation, but would also dramatically increase the computation costs when evaluating on new data. For knowledge bases, it is possible to leverage domain expertise and select an external knowledge base that would be most helpful for the domain adaptation of factuality evaluation. It is also possible to leverage a combination of existing knowledge bases for FACTKB's factuality pretraining, while the specific combination and how to apply different factuality pretraining to different KBs are hard to determine. All in all, FACTKB presents a general KB-enhanced factuality metric with numerous possibilities, while we leave some of these considerations to future work.

**FACTKB training is not end-to-end.**   FACTKB has a two-step training process: pretraining with KB-based factuality pretraining and fine-tuning on factuality evaluation datasets. This creates several limitations, among which is the difficulty of hyperparameter tuning. Appendix 3.1 presents the study of hyperparameters in the factuality pretraining stage, which demonstrates that FACTKB works best with a moderate but not excessive amount of factuality pretraining. This reliance on certain hyperparameter configurations is further complicated by more hyperparameter choices in the fine-tuning stage. While the current hyperparameter setting in Appendix 3.1 achieves state-of-the-art empirical performance, we acknowledge the difficulty in FACTKB hyperparameter tuning.

**Out-of-Domain Factuality Evaluation.**   An important focus of this work is out-of-domain factuality evaluation: Summarization systems face input documents from varying domains, which requires factuality metrics to also generalize to different document domains. Existing metrics struggle with semantic frame errors and such struggle is exacerbated by the domain shift of entities and relations, while FACTKB offers a stronger and more generalizable factuality metric. However, in this work, we mainly focused on the additional domain of scientific literature, while other potential domains remain underexplored such as social media (Syed et al., 2019; Kano et al., 2018; He et al., 2020). We leave it to future work the exploration of FACTKB and existing factuality metrics on more document domains that are present in summarization systems.

**Tradeoff between Performance and Granularity** Existing approaches (Kryściński et al., 2020; Takatsuka et al., 2022) struggle with semantic frame errors and involve heavy preprocessing, while they provide fine-grained analysis and specific localization of summarization errors. FACTKB achieves significantly better factuality evaluation results and is easier to use while lacking the ability of error localization. We argue that this tradeoff should be considered with the use case in mind: for LLM evaluation, it is better to have an accurate metric for benchmarking efforts and an efficient metric for handling large-scale LM generation. As a result, FACTKB provides a valuable tool for factuality evaluation and LLM research.

## Ethics Statement

**LM and KB Bias**   FACTKB is initialized with pretrained language model checkpoints and leverages knowledge-base-based factuality pretraining. Consequently, FACTKB might pick up the biases of the adopted language models (Liang et al., 2021; Nadeem et al., 2021; Shaikh et al., 2023; Tan and Celis, 2019) and knowledge bases (Fisher et al., 2020; Mehrabi et al., 2021). As a result, FACTKB might leverage these biases in judging the factuality of summaries, further reinforcing the bias in text summarization systems. We leave it to future work on understanding and mitigating the bias of factuality metrics.

**Misuse Potential**   FACTKB leverages high-quality and factual knowledge bases to generate factuality pretraining corpora and augment LM's ability to stay factual with respect to entities and relations discussed in the summary and document. On the contrary, if non-factual and misleading knowledge is leveraged for the three factuality pretraining strategies, it might jeopardize the factuality of FACTKB and make it insensitive to misinformation and falsehoods in summaries and documents. As a result, we encourage the responsible use of FACTKB and the factuality pretraining methodology.

## Acknowledgements

We thank the reviewers, the area chair, members of Tsvetshop, and the UW NLP Group for their feedback. This research was supported by This research is supported in part by the Office of the Director of National Intelligence (ODNI), Intelligence Advanced Research Projects Activity (IARPA), via the HIATUS Program contract #2022-22072200004. This material is also funded by the DARPA Grant under Contract No. HR001120C0124. We also gratefully acknowledge support from NSF CAREER Grant No. IIS2142739 and the Alfred P. Sloan Foundation Fellowship. The views and conclusions contained herein are those of the authors and should not be interpreted as necessarily representing the official policies, either expressed or implied, of ODNI, IARPA, or the U.S. Government. The U.S. Government is authorized to reproduce and distribute reprints for governmental purposes notwithstanding any copyright annotation therein.

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

## A Merging the three strategies

We also tried combining the three factuality pre-training strategies to obtain FACTKB-COMBINED. We evaluate it on the FactCollect dataset and present results in Table 7. It is demonstrated that FACTKB-COMBINED is not significantly better than using a single factuality pretraining strategy, while we will make all versions of FACTKB publicly available.

## B Qualitative Analysis

We present examples of (summary, article) pairs and their factuality scores in Table 9 and 10, where FACTKB is significantly closer to human judgment than existing factuality metrics. It is demonstrated that while existing factuality metrics are insensitive to major errors in entities and relations, FACTKB is capable of identifying inconsistencies and enforcing strict factuality standards.

## C Dataset Details

We present more details about the adopted datasets in Table 8. There might be minor differences in certain numbers with the original dataset as a result of data preprocessing. FRANK (Pagnoni et al., 2021) does not explicitly have binary labels such as {FACTUAL, NOT FACTUAL}. It also does not have a training set due to its nature as an evaluation benchmark. HealthVer (Sarrouti et al., 2021) and SciFact (Wadden et al., 2020) originally had NOT ENOUGH INFORMATION labels, while we removed such examples in the out-of-domain factuality evaluation to ensure their compatibility with FactCollect.

## D Baseline Details

We present more details about baseline factuality metrics in the following:

- **BERTScore** (Zhang et al., 2019) is a general metric for text generation evaluation based on pretrained BERT (Devlin et al., 2019).

- **QAGS** (Wang et al., 2020a) is a QA-based factuality metric, asking questions about summaries and articles while examining whether the answers are consistent.

- **QUALS** (Nan et al., 2021) is a QA-based factuality metric that uses QAGen (Shakeri et al., 2020) to generate both questions and answers from the summary.

- **DAE** (Goyal and Durrett, 2020) leverages the dependency structure of the summary and article to design a factuality metric.

- **SummaC** (Laban et al., 2022) proposes to revisit and repurpose NLI models for detecting factual inconsistencies in text summarization.

- **FalseSum** (Utama et al., 2022) augments NLI training data with controllable text generation for better factuality evaluation.

- **FactCC** (Kryściński et al., 2020) is an entailment-based factuality metric trained on synthetic data evaluating factuality with binary classification. **FactCC+** is a variant of FactCC providing explanations. **FactCC+** is an enhanced version trained with human-annotated data.

- **FactGraph** (Ribeiro et al., 2022) is an entailment-based factuality metric based on jointly analyzing the textual content and AMR graphs of the summary and article. **FactGraph-adapters** is an enhanced version with pretrained adapters for both the text and graph modules.

## E LM and KB Details

In Section 5.2, we explored whether FACTKB is compatible with different language models and knowledge bases. For LMs, we used the ROBERTA-BASE, GOOGLE/ELECTRA-BASE-DISCRIMINATOR, FACEBOOK/BART-BASE, ALBERT-BASE-V2, MICROSOFT/DEBERTA-V3-BASE, DISTILROBERTA-BASE LM checkpoints on Huggingface Transformers. For the six KBs, we used their organized versions: YAGO15k at Lacroix et al. (2019), Wikidata5M at Wang et al. (2021), Atomic at West et al. (2022), ConceptNet at Zhang et al. (2022b), KGAP at Feng et al. (2021), and UMLS at Zhang et al. (2022b).

## F Statistical Significance Test Details

We use the student $t$-test for statistical significance analysis throughout the paper. Specifically, the $t$-test calculator for 2 independent means [3] was adopted for the calculations. We use (*) to denote statistical significance in Tables 2 and 5.

---

[3]https://www.socscistatistics.com/tests/studenttest/default2.aspx

| Model | All Data | | CNN/DM | | XSUM | |
|---|---|---|---|---|---|---|
| | **BACC** | **F1** | **BACC** | **F1** | **BACC** | **F1** |
| FACTKB-WIKI | 89.3 (±0.4) | 89.5 (±0.5) | 77.3 (±0.3) | 88.2 (±0.6) | 77.3 (±1.3) | 91.8 (±1.2) |
| FACTKB-EVIDENCE | 89.4 (±0.2) | 89.5 (±0.3) | 77.7 (±1.4) | 87.9 (±0.7) | 76.8 (±1.9) | 90.8 (±0.8) |
| FACTKB-WALK | 89.1 (±0.4) | 89.3 (±0.5) | 78.3 (±1.2) | 87.7 (±0.4) | 76.4 (±0.3) | 90.4 (±1.4) |
| FACTKB-COMBINED | 89.0 | 89.7 | 76.0 | 88.1 | 74.2 | 89.1 |

Table 7: Performance of various FACTKB settings on the FactCollect dataset.

| Dataset | # Datapoint | # Class | Class Distribution | Train/Dev/Test Split | Proposed In |
|---|---|---|---|---|---|
| FACTCOLLECT | 9,567 | 2 | 4994 / 4573 | 8667 / 300 / 600 | Ribeiro et al. (2022) |
| FRANK | 2,246 | / | / | 0 / 671 / 1575 | Pagnoni et al. (2021) |
| COVIDFACT | 1,257 | 2 | 401 / 856 | 846 / 94 / 317 | Saakyan et al. (2021) |
| HEALTHVER | 4,447 | 3 → 2 | 2,758 / 1,689 | 3,340 / 508 / 599 | Sarrouti et al. (2021) |
| SCIFACT | 773 | 3 → 2 | 508 / 265 | 508 / 56 / 209 | Wadden et al. (2020) |

Table 8: Statistics of the datasets and benchmarks adopted in this work.

## G  Computational Resources

We used a GPU cluster with 16 NVIDIA A40 GPUs, 1988G memory, and 104 CPU cores for the experiments. Factuality pretraining with the default hyperparameters takes around 1.5 hours, while fine-tuning language models on the FactCollect dataset takes around 30 minutes.

## H  Scientific Artifacts

FACTKB would not be possible without many open-source scientific artifacts, including pytorch (Paszke et al., 2019), pytorch lightning (Falcon and The PyTorch Lightning team, 2019), transformers (Wolf et al., 2020), sklearn (Pedregosa et al., 2011), numpy (Harris et al., 2020), nltk (Bird et al., 2009), and the six adopted knowledge bases (Pellissier Tanon et al., 2020; Vrandečić and Krötzsch, 2014; West et al., 2022; Speer et al., 2017; Feng et al., 2021; Zhang et al., 2022b). We commit to making our code and data publicly available upon acceptance to facilitate reproduction and further research.

| QAGS | DAE | FactCC | FactKB | Gold | Summary | Article |
|---|---|---|---|---|---|---|
| 0.3000 | 0.9990 | 1.0000 | 0.0035 | 0 | plans to build a new generation of royal navy frigates on the isle of wight have been submitted to the government. | The decommissioned Type 22 frigates HMS Cumberland, HMS Campbeltown, HMS Chatham and HMS Cornwall are currently moored in Portsmouth Harbour.Bidders had until 23 January to register an interest in the former Devonport-based ships.The BBC understands no proposals to preserve the ships have been submitted.Those who have registered an interest are finalising their bids with viewings set to take place in late February and March.A final decision is not expected until the spring.The government's Disposal Services Authority, which is handling the sale, wants to award at least one of the frigates to a UK ship recycler to determine the capacity of the UK's industry in the field.Penny Mordaunt, Conservative MP for Portsmouth North, said it was important UK recyclers had the chance to prove themselves in the field but she was also keen to see at least one of them saved from the scrapyard.She added: "For anyone that has served on a ship it's your home, you've literally been through the wars with it... and you want them to have a noble second life."My preference is to go for the reef and diving attraction."We've got to get best value for the budget but a reef would also generate income for part of the country through tourism."The Ministry of Defence has previously said it will "consider all options" for the frigates to ensure "best financial return for the taxpayer".A spokeswoman would not comment on the number or nature of the bids received due to "commercial sensitivity".Originally designed as a specialist anti-submarine ship, the Type 22 frigate evolved into a powerful surface combatant with substantial anti-surface, anti-submarine and anti-aircraft weapons systems.They were also known for having excellent command and control, and communication facilities, making them ideal flagships on deployments, with a complement of about 280 crew.Last year, the aircraft carrier HMS Ark Royal was sold as scrap for £3m. |
| 0.5333 | 0.9296 | 1.0000 | 0.0043 | 0 | an elephant has been hit by a stone at a zoo in western france after it was hit by a tree. | The stone got past the elephant's fence and a ditch separating the animal and visitors, the zoo said in a statement.The girl was taken to hospital and died within a few hours, the zoo added.The zoo statement said the enclosure met international standards and said "this kind of accident is rare, unpredictable and unusual".Africa Live: More on this and other storiesThe statement went on (in French) to point out two other recent incidents in the US:Phyllis Lee, Scientific Director of the Amboseli Trust for Elephants, says that targeted throwing of stones and branches by elephants is very unusual."It can happen when elephants are frustrated or bored. In my opinion, it's unlikely the elephant was directly targeting the girl - but exhibiting frustration. You can't predict what animals in captivity will do."The moments after the girl was struck at Rabat Zoo on Tuesday were filmed by a bystander and uploaded onto YouTube.The video shows the elephant waving its trunk behind a fence and swerves round to show a stone on the ground.Metres away people are gathered around the girl, holding her head and stroking her leg. |
| 0.6000 | 0.9994 | 1.0000 | 0.0037 | 0 | a woman has been arrested after a fire broke out in a restaurant in greater manchester city centre, police have said. | The victim was queuing for food at the branch in St George's Street, Canterbury at about 02:15 GMT on Friday when the assault occurred.Investigating officers said three men entered the restaurant and began being noisy and bumping into people.It is believed one of the group then set light to the woman's hair.Officers have released CCTV images of three men they are keen to speak to regarding the attack.Det Sgt Barry Carr said: "Fortunately the fire was put out quickly and the victim was not seriously hurt, but things could clearly have turned out much worse."This was a nasty and extremely dangerous thing to do, and I urge anyone who recognises the men in the CCTV images to contact me as soon as possible." |
| 0.8000 | 0.9974 | 1.0000 | 0.0044 | 0 | tata steel has confirmed it is in talks with the company about selling its long products division. | The firm said it had signed a Letter of Intent to enter into exclusive negotiations with Liberty House Group.More than 1,700 people are employed in the division, which has factories in Rotherham and Stocksbridge.Steel union Community said it welcomed news of negotiations following "months of unnecessary stress and concern".More on this and other South Yorkshire storiesThe union's general secretary Roy Rickhuss said: "This is a positive step for the UK steel industry; however there remain huge challenges which government must address."The union said it would be seeking urgent talks with Liberty House Group and would be asking what their plans were for investment, protecting jobs and providing decent pensions for members in retirement.Tata Steel's UK boss Bimlendra Jha said the announcement was "an important step forward"."We now look forward to working with Liberty on the due diligence and other work streams so that the sale can be successfully concluded," he said.The Speciality Steels unit makes high-end components for the automotive, aerospace and oil industries.In April, Tata sold its long-products division, based in Scunthorpe, to Greybull Capital, a UK-based investment firm. |
| 0.3000 | 0.9990 | 1.0000 | 0.0058 | 0 | the site of a new burial site in oxford has been approved by the city council. | Oxford City Council said the money had mostly been used for "ground investigations of possible sites" but nowhere suitable had been found.Two cemeteries still have space, in Wolvercote and Botley, but they are expected to be full by 2018 and 2021.The council said it had not given up and was "still exploring options".Linda Smith, board member for leisure, parks and sport, said the council has been "searching for a suitable new burial site for many years".She added: "But ultimately, as with new housing sites, we have run out of suitable land within Oxford."So far all the council-owned sites that we have identified have, following ground investigations and surveys, had to be discounted."Either due to the size of the site, the ground conditions, a high water table or a covenant restricting the use of the site."After the two remaining cemeteries are full the council said only the reopening of family plots, the use of a few reserved plots, and the interment of ashes would be possible.The last increase in burial space in Oxford was in 1932. |

Table 9: Qualitative analysis of FACTKB and existing factuality metrics, part 1.

| QAGS | DAE | FactCC | FactKB | Gold | Summary | Article |
|---|---|---|---|---|---|---|
| 0.6000 | 0.9886 | 1.0000 | 0.0037 | 0 | plans to demolish and demolish parts of a seaside resort and build more than 1, 000 old buildings have been approved. | Three Victorian hotels will go to make way for a six-storey, four star hotel and two assisted-living apartment blocks, at East Cliff in Bournemouth.English Heritage strongly objected to the scale of the development in what is a designated conservation area.But, councillors voted seven to three in favour saying it would help tourism.Chair of the planning board and Conservative ward councillor David Kelsey, said the buildings earmarked for demolition were nice but no longer "necessarily functional"."They've come to the end of their working lives, we need to preserve the tourism aspect while improving living for older people in the town," he said."The loss of buildings and trees are always regrettable but we can't stand still, we need to move forward."The site on Grove Road and East Overcliff Drive will get a 90-room hotel along with a nine-storey and seven-storey building, comprising 122 assisted-living apartments.Applicants The East Cliff Project LLP will demolish Bay View Court, The Cottonwood and the Ocean View hotels.The council received 246 letters supporting the plans.Forty-nine residents and the Ancient Monuments Society wrote to object to the demolition, stating that despite being altered, they still "give a sense of the historic character of the area".English Heritage said the scale of the development would cause "severe harm" to the conservation area. |
| 0.6000 | 0.9928 | 1.0000 | 0.0059 | 0 | russiaś new president has called for a new law to allow russian citizens to be barred from leaving the country. | Pro-Kremlin party A Just Russia put forward both bills, and linked them directly to the situation in Ukraine.Separatist and pro-Russian feelings are strong in Ukraine's Crimea region, which is now the focus of the crisis.Russian MPs say a referendum or a plea from a territory's leaders would be enough to trigger the new provisions.There are already many Russian citizens in Crimea.In Sevastopol, base of the Russian Black Sea Fleet, a majority hold Russian passports.Under Russia's existing law, a neighbouring state would have to sign a treaty with Russia to allow part of its territory to become a new "subject" of the Russian Federation.But Mikhail Yemelyanov, deputy leader of A Just Russia, said the law had been drafted for peaceful times, and did not go far enough for situations where a state was falling apart."In conditions where a neighbouring state is disintegrating I don't think the Russian Federation should be restricted in its ability to accept a territory whose people have expressed a clear will and desire to be in Russia," he said.Since Russia's war with Georgia in 2008, the breakaway Georgian territories of Abkhazia and South Ossetia have come under Moscow's control.Russia poured troops into both regions to help pro-Russian separatists who did not recognise Georgia's authority.The other bill to be considered by the Duma - Russia's lower house - would speed up the procedures for issuing Russian passports.Passport applicants would not have to pay a state tax, and previous residence in Russia would no longer be required.In addition, they would not have to have sufficient funds to support themselves and would not have to give up their Ukrainian citizenship.The bill's preamble says it is aimed "at supporting the fraternal people of Ukraine, especially the Russian-speaking ones, who are defenceless in the face of the 'brown threat'," a reference to World War Two fascists who wore brown uniforms.The bill would allow Ukrainians to apply for Russian passports at Russian diplomatic missions before 1 August, and they could become citizens after two months, instead of waiting a year, as is currently the norm.The plan to have a new fast-track procedure for issuing Russian passports was announced in Sevastopol on Thursday by A Just Russia leader Sergei Mironov.Several Russian MPs have also gone to Crimea, including Russian celebrities - former Olympic ice skating champion Irina Rodnina, former cosmonaut Valentina Tereshkova and heavyweight boxer Nikolai Valuev. |
| 0.7000 | 0.9984 | 1.0000 | 0.0047 | 0 | a 19-year-old man has been arrested in connection with the fatal shooting of an 18-year-old student in the southern indian state of | The shooting occurred at a hostel attached to the private Pragati Residential School in Bangalore city.Police say the alleged gunman, identified as Mahesh, was working as an office assistant in the school.Incidents of gun crime at schools and colleges in India are very rare. It is not clear what prompted the shooting.Police said on Thursday that Mahesh had been remanded until 12 April.Mahesh is alleged to have barged into the room of 18-year-old Gautami and shot her in the head with a pistol on on Tuesday evening.He then shot another student, Sirisha, who suffered severe injuries but is believed to be out of danger, say police.He was arrested on Wednesday after a manhunt.India has strict control laws, although a large number of feuds are settled with firearms.In 2007, a 14-year-old schoolboy was shot dead by two fellow students at a school campus near the capital, Delhi. |
| 0.5000 | 0.9995 | 1.0000 | 0.0036 | 0 | police have appealed for help to trace two men who threatened a woman with a knife at a quarry in fife. | The men entered the Post Office in Quarrywood Avenue, in the Barmulloch area, at 07:55 on Friday.They threatened a member of staff with a knife and demanded money before escaping with the cash.The 27-year-old worker was said by police to have been badly shaken but otherwise unharmed by the ordeal.Both suspects are white, and one of them was about 35-40 years old with short brown hair and wearing a black jumper.Det Sgt Raymond Hunter said officers had been carrying out door-to-door inquiries and were in the process of collecting CCTV images from the surrounding area.He added: "There are a number of other shops in this area and people may have seen the two men prior to or after the incident."I am therefore appealing to anyone who was in the area or any local residents to contact us - any information you have could assist our enquiry." |
| 0.5000 | 0.9999 | 1.0000 | 0.0063 | 0 | the number of people using plastic carrier bags in england has reached a record high. | The Department for Environment, Food and Rural Affairs found the number had gone up by 200 million since 2013.There has been a big problem with plastic carrier bags in the last few years, many of them can't be recycled and are often thrown away after they have been used.The bags end up in rubbish dumps and even rivers causing big problems for the environment.From October people in England will have to pay 5p for their plastic bags in a bid to encourage them to reuse the ones that they already have.Supermarkets in Wales, Scotland and Northern Ireland, where people are charged for carrier bags, have all seen a decrease in bags used.Campaigners are hoping the charge in England will lessen the amount of bags being thrown away, helping the environment. |

Table 10: Qualitative analysis of FACTKB and existing factuality metrics, part 2.