# OpenReview forum: "FactKB: Generalizable Factuality Evaluation using Language Models Enhanced with Factual Knowledge"
_EMNLP/2023/Conference — EMNLP 2023 Main_

### Official Review · Reviewer_BoeU · 2023-08-01

**Soundness:** 4

**Excitement:**

4: Strong: This paper deepens the understanding of some phenomenon or lowers the barriers to an existing research direction.

**Missing References:**

Ranking Generated Summaries by Correctness: An Interesting but Challenging Application for Natural Language Inference (Falke et al., ACL 2019)

**Paper Topic And Main Contributions:**

This paper studies the evaluation of factual consistency in automatic summarization. It discusses the challenges of evaluating factual consistency, and proposes a new approach that uses an external knowledge base (KB) to pre-train a classification model to evaluate factual inconsistencies in summaries.

In particular, it introduces three pretraining objectives that leverages connections/relations between entities in the KG which are based on (i) entity-specific facts (ii) facts aligned with relevant evidence – e.g. a paragraph – and (iii) random walks with traverse the KG in a multi-hop setting. The model is pretrained using an encyclopedic KG based on Wikidata.

The model is further fine-tuned in a human-curated dataset for factual consistency assessment and evaluated in-domain (news summarization) and on out-of-domain (scientific literature) datasets. The results indicate that the model improves over previous work for factual consistency evaluation, improving correlations with human annotations and metrics such as BACC and F1.

The paper further investigates the performance of the method using other KGs indicating that such pretraining can help to improve factuality evaluation compared to vanilla pretrained models.

**Questions For The Authors:**

Could the authors provide some examples of semantic frame errors that the model is able to capture (e.g., from Frank benchmark) where other systems (such as FactGraph-edge) fails in capture? There are some examples of evaluations in the Appendix but they are not fine-grained explained.

**Reasons To Accept:**

- The paper leverages the structure of external KB for proposing three factuality pretraining objectives: entity wiki, evidence extraction, and knowledge walk, with the goal of enabling model capabilities such as understanding multi-hop claims and evaluate facts from evidences.
- The paper provides improvements over existing factuality evaluation on in- and out-of-domain settings.
- While previous approaches such as FactGraph and DAE rely on parsers as an additional step for evaluation (which increases the inference complexity), the model does not require additional steps which makes the inference easier and faster (but required a pretraining step).

**Reasons To Reject:**

- Differently to previous approaches such as FactGraph-edge and DAE, the proposed method is not able to detect fine-grained factuality assessment which is an important feature to indicate specific factuality issues compared to a binary label for the entire summary.
- The paper focuses on improvements on semantic frame errors (related to entities, events and their relations) but lacks a deeper exploration of this improvement. Some use cases / qualitative analysis could bring more insights to be discussed.

**Reproducibility:**

4: Could mostly reproduce the results, but there may be some variation because of sample variance or minor variations in their interpretation of the protocol or method.

**Reviewer Confidence:**

4: Quite sure. I tried to check the important points carefully. It's unlikely, though conceivable, that I missed something that should affect my ratings.

---

> ### Author Rebuttal · Authors · 2023-08-29
>
> > Differently to previous approaches such as FactGraph-edge and DAE, the proposed method is not able to detect fine-grained factuality assessment which is an important feature to indicate specific factuality issues compared to a binary label for the entire summary.
>
> We refer the Reviewer to the “Tradeoff between Performance and Granularity” section in the limitations discussion. In short, FactKB aims at presenting an efficient and low-burden approach for factuality evaluation, which is especially valuable in the context of large-scale generated text from large language models. Consequently, the interpretability aspect is not emphasized in FactKB by design. FactKB can be used in combination with more interpretable approaches to factuality evaluation [1-2], which are much more compute-intensive.
>
> FactKB and more interpretable approaches could also be combined to form a pipeline: given huge quantities of machine-generated text, we could first use the much more efficient FactKB for a round of quick and large-scale detection and evaluation, while leaving the more nuanced instances with low confidence scores for enhanced and interpretable analysis. This pipeline would bring out the best of both worlds, while we leave it for future work.
>
> [1] Kryściński, Wojciech, et al. "Evaluating the Factual Consistency of Abstractive Text Summarization." EMNLP 2020.
>
> [2] Takatsuka, Masato, Tetsunori Kobayashi, and Yoshihiko Hayashi. "Phrase-Level Localization of Inconsistency Errors in Summarization by Weak Supervision." COLING 2022.
>
>
> > The paper focuses on improvements on semantic frame errors (related to entities, events and their relations) but lacks a deeper exploration of this improvement. Some use cases / qualitative analysis could bring more insights to be discussed.
>
> > Could the authors provide some examples of semantic frame errors that the model is able to capture (e.g., from Frank benchmark) where other systems (such as FactGraph-edge) fails in capture? There are some examples of evaluations in the Appendix but they are not fine-grained explained.
>
> Consider the following example:
>
> **Summary**: an elephant has been hit by a stone at a zoo in western france after it was hit by a tree.
>
> **Document**: The stone got past the elephant’s fence and a ditch separating the animal and visitors, the zoo said in a statement.The girl was taken to hospital and died within a few hours, the zoo added.The zoo statement said the enclosure met international standards and said "this kind of accident is rare, unpredictable and unusual".Africa Live: More on this and other stories. The statement went on (in French) to point out two other recent incidents in the US:Phyllis Lee, Scientific Director of the Amboseli Trust for Elephants, says that targeted throwing of stones and branches by elephants is very unusual."It can happen when elephants are frustrated or bored. In my opinion, it’s unlikely the elephant was directly targeting the girl - but exhibiting
> frustration. You can’t predict what animals in captivity will do."The moments after the girl was struck at Rabat Zoo on Tuesday were filmed by a bystander and uploaded onto YouTube.The video shows the elephant waving its trunk behind a fence and swerves round to show a stone on the ground.Metres away people are gathered around the girl, holding her head and stroking her leg.
>
> **Analysis**: While it was the girl who was hit by a stone, the summary incorrectly claimed that the elephant was hit by a stone, confusing the entities and relations between stone, girl and elephant and making a semantic frame error. While existing approaches (QAGS, DAE, FactCC, FactGraph) evaluate this summary as factual (giving > 0.5 factuality scores), FactKB gives a score of 0.0035, indicating that the summary is not factual.
>
> We will include a qualitative analysis section in the additional page, dedicated to fine-grained analysis of summaries and documents such as the above example and how FactKB improves over existing models.
>
> > Ranking Generated Summaries by Correctness: An Interesting but Challenging Application for Natural Language Inference (Falke et al., ACL 2019)
>
> We will include this work in the related work discussion.
>
> We thank the reviewer for their insightful questions. We will include the above clarifications and a qualitative analysis section in the final version of the paper.

---

### Official Review · Reviewer_S3Ep · 2023-08-02

**Soundness:** 4

**Excitement:**

3: Ambivalent: It has merits (e.g., it reports state-of-the-art results, the idea is nice), but there are key weaknesses (e.g., it describes incremental work), and it can significantly benefit from another round of revision. However, I won't object to accepting it if my co-reviewers champion it.

**Paper Topic And Main Contributions:**

This paper studies on evaluating the factual consistency of automatically generated summaries. Previous factuality classifiers are not robust to ever-growing information, where the distribution of entities, events, and relations vary greatly across time and domains. To address these issues, this paper develops a novel factuality evaluation model FACTKB with improved factual knowledge representation, focusing on entities and relations. To be more concrete, FACTKB is built on LMs augmented with factual knowledge (Section 2).

**Questions For The Authors:**

1. Evidential fact verification tasks are also research hotspots. Is it possible to use a single model (e.g., evidence extraction, instead of three strategies)  for the current task? By the way, the motivations of Entity and Knowledge walk are not so clear.
2. Can you give interpretability of the model, e.g, presenting examples to illustrate which tokens or sentences are selected or assigned more attention weights for factuality evaluation?
3. According to Table 3, 4, it seems that FACTKB-EVIDENCE is not so effective. Maybe you can neglect FACTKB-EVIDENCE?

**Reasons To Accept:**

1. This paper is well-organized and well-written.
2. This paper offeres a comprehensive analysis of limitations.
3. This paper proposes three strategies of factuality pretraining: Entity Wiki, Evidence Extraction, KnowledgeWalk.
4. The descriptions of the proposed model and training process are clear and easy to understand.
5. This paper considers both in-domain evaluation and generalizable factuality evaluation.

**Reasons To Reject:**

1. The proposed method is somewhat simple and coarse-grained, and lacks interpretability.
2. The experimental analysis is expected to be more detailed.

**Reproducibility:**

4: Could mostly reproduce the results, but there may be some variation because of sample variance or minor variations in their interpretation of the protocol or method.

**Reviewer Confidence:**

4: Quite sure. I tried to check the important points carefully. It's unlikely, though conceivable, that I missed something that should affect my ratings.

---

> ### Author Rebuttal · Authors · 2023-08-29
>
> > Evidential fact verification tasks are also research hotspots. Is it possible to use a single model (e.g., evidence extraction, instead of three strategies) for the current task? By the way, the motivations of Entity and Knowledge walk are not so clear.
>
> The three proposed strategies are orthogonal approaches that could each be used individually for factuality evaluation. In addition, the three strategies have respective pros and cons, evident in the analysis in Figures 3 and 4. The entity wiki strategy is motivated by the task of knowledge graph completion which helps models understand relationships between entities. We discuss this in detail in lines 123-132 for elaboration. The knowledge walk strategy is motivated by the need for multi-hop knowledge reasoning in understanding factuality which enables models to reason across multiple entity chains.  We elaborate this in lines 174-182.
>
> > Can you give interpretability of the model, e.g, presenting examples to illustrate which tokens or sentences are selected or assigned more attention weights for factuality evaluation?
>
> We refer the Reviewer to the “Tradeoff between Performance and Granularity” section in the limitations discussion. In short, FactKB aims at presenting an efficient and low-burden approach for factuality evaluation, which is especially valuable in the context of large-scale generated text from large language models. Consequently, the interpretability aspect is not emphasized in FactKB by design. FactKB can be used in combination with more interpretable approaches to factuality evaluation [1-2], which are much more compute-intensive.
>
> In addition, whether the attention weights from a neural language model such as RoBERTa could be considered “interpretability” is subject to debate [3-4], thus we did not include such results and did not claim for interpretability.
>
> [1] Kryściński, Wojciech, et al. "Evaluating the Factual Consistency of Abstractive Text Summarization." EMNLP 2020.
> [2] Takatsuka, Masato, Tetsunori Kobayashi, and Yoshihiko Hayashi. "Phrase-Level Localization of Inconsistency Errors in Summarization by Weak Supervision." COLING 2022.
>
> [3] Jain, Sarthak, and Byron C. Wallace. "Attention is not Explanation." NAACL 2019.
>
> [4] Wiegreffe, Sarah, and Yuval Pinter. "Attention is not not Explanation." EMNLP 2019.
>
> > According to Table 3, 4, it seems that FACTKB-EVIDENCE is not so effective. Maybe you can neglect FACTKB-EVIDENCE?
>
> Empirically, FactKB-evidence is actually the best-performing model on the FactCollect benchmark in Table 2 (the All data columns). Indeed, it does not perform most strongly in the settings of Tables 3 and 4, showing that there are advantages and disadvantages of all three strategies, which is why we included all the results to best inform readers. Conceptually, FactKB-evidence enables the integration of external information such as Wikipedia and retrieval corpora to augment language models for factuality evaluation. As a result, we kept the FactKB-evidence approach in the paper.
>
> We thank the reviewer for their insightful questions. We will include the above clarifications in the final version of the paper.

---

### Official Review · Reviewer_ttnU · 2023-08-04

**Typos Grammar Style And Presentation Improvements:** 1. In Figure 3, is the color used to …
**Soundness:** 4

**Excitement:**

3: Ambivalent: It has merits (e.g., it reports state-of-the-art results, the idea is nice), but there are key weaknesses (e.g., it describes incremental work), and it can significantly benefit from another round of revision. However, I won't object to accepting it if my co-reviewers champion it.

**Paper Topic And Main Contributions:**

The objective of this paper is to address the issue of factuality detection, which involves determining whether a summary generated from an article accurately consistent with the original content. Previous research has highlighted that a significant error in factuality evaluation is related to semantic frame errors, specifically concerning entities and their relationships.

To address this, the main contribution of this paper is the proposal of FactKB, containing three pre-training objectives aimed at improving a model's ability to better represent entities and relationships. These objectives involve further training the pre-trained language models using external knowledge graphs such as Yago by prediction missing entities in a triplet or random walk path. The experimental results demonstrate that models trained on external knowledge graphs exhibit enhanced performance on the overall fact validation dataset, particularly in relation to the semantic frame type.

**Questions For The Authors:**

A: Were the baseline results in Tab2-Tab4 obtained directly from the reported results in previous research? If so, it is necessary to indicate this in the report.

B: Regarding the significant test in Tab2 and Tab4, was it conducted to compare Vanilla RoBERTa with FactKB-*? Additionally, were there any significant differences observed among the three models: FactKB-Wiki, FactKB-Evidence, and FactKB-Walk?

C: In Appendix B, it is mentioned that combining the three strategies did not yield better results than using single strategies. Could you provide any explanations for this finding, considering that each of the three strategies has its own specific focus?

D:  In Figure 3, how was FactKB-evidence being trained on other knowledge graphs that are not aligned with Wikipedia? How are the training data constructed?

**Reasons To Accept:**

This paper conducts robust experiments, which include three pre-training objectives and an examination of compatibility across multiple knowledge graphs.

It provides a fine-grained analysis, demonstrating that FactKB mitigates semantic frame errors. Furthermore, the paper reveals that FACTKB enhances performance on both in-domain and out-of-domain scientific datasets.

**Reasons To Reject:**

The paper shows comprehensive experimental results, but it lacks sufficient emphasis on highlighting the differences between FactKB and other existing models. Particularly, it needs to provide more details about the experimental setup, such as the training data used for each baseline model, to convince readers that the comparison is fair. This lack of comparison to existing methods can be considered a weakness and a reason for potential rejection.

**Reproducibility:**

4: Could mostly reproduce the results, but there may be some variation because of sample variance or minor variations in their interpretation of the protocol or method.

**Reviewer Confidence:**

3: Pretty sure, but there's a chance I missed something. Although I have a good feel for this area in general, I did not carefully check the paper's details, e.g., the math, experimental design, or novelty.

---

> ### Author Rebuttal · Authors · 2023-08-29
>
> > Particularly, it needs to provide more details about the experimental setup, such as the training data used for each baseline model, to convince readers that the comparison is fair.
>
> Below we provide details on our baselines and experimental setup. We will expand the experimental setup in the baseline paragraph of Section 3 in the revised version to include these details.
>
> In Table 2, FactKB and baselines are first pre-trained on respective synthetic data (if any), then fine-tuned on the FactCollect training set, and finally evaluated on the FactCollect test set. We report results on the full test set (all data) and the two subsets (CNN/DM and XSUM).
>
> In Table 3, the above trained models are evaluated on the test set of the FRANK benchmark.
>
> In Table 4, FactKB and baselines are first pre-trained on respective synthetic data (if any), then fine-tuned on the training sets of CovidFact, HealthVer, or SciFact, and then evaluated on their respective test splits.
>
> In Figure 4 and Table 5, we use the FactKB and baselines trained on the FactCollect benchmark (Table 2) and evaluate with the official FRANK script.
>
> In short, the above experimental setups and training/test data splits are designed for fair comparisons against strong baselines.
>
> > Were the baseline results in Tab2-Tab4 obtained directly from the reported results in previous research? If so, it is necessary to indicate this in the report.
>
> For Tables 2 and 3, the results for QAGS, QUALS, FactCC, FactCC+, FactGraph, and FactGraph-adapters are taken from the FactGraph work [1] as the datasets and splits (Fact collect and FRANK) are identical. For other baselines in Tables 2 and 3 and all of Table 4, we conducted experiments to evaluate existing model checkpoints if available (e.g., FactCC and FactGraph) or train from scratch.
>
> [1] Ribeiro, Leonardo, et al. "FactGraph: Evaluating Factuality in Summarization with Semantic Graph Representations." NAACL 2022.
>
> > Regarding the significant test in Tab2 and Tab4, was it conducted to compare Vanilla RoBERTa with FactKB-*? Additionally, were there any significant differences observed among the three models: FactKB-Wiki, FactKB-Evidence, and FactKB-Walk?
>
> The significance tests and asterisks indicate whether the FactKB variant significantly outperforms the best-performing baseline model. In some experiments, we observed statistical differences among the three FactKB variants, such as FactKB-walk significantly outperforming the other two on CNN/DM in Table 2 in terms of BACC scores, but in most experiments, they all have similar performance.
>
> > In Appendix B, it is mentioned that combining the three strategies did not yield better results than using single strategies. Could you provide any explanations for this finding, considering that each of the three strategies has its own specific focus?
>
> We hypothesize that this is because the advantages from the different knowledge reasoning augmentations could be more nuanced and therefore on a general benchmark like FactCollect their difference is marginal (as similarly indicated in Table 2). However, in more reasoning-intensive factuality scenarios like fact-checking complex claims [1], combining the three strategies might yield greater improvement over individual approaches, which we leave for future work.
>
> [1] Pan, Liangming, et al. "Fact-Checking Complex Claims with Program-Guided Reasoning." arXiv preprint arXiv:2305.12744 (2023).
>
> > In Figure 3, how was FactKB-evidence being trained on other knowledge graphs that are not aligned with Wikipedia? How are the training data constructed?
>
> For knowledge graphs other than Wikipedia, we continue to use the Wikipedia Search API to identify entities that are linkable to Wikipedia entries and use its Wikipedia description to construct the training data. The success rate for entity linking varies from 62% to 98% for the five other knowledge graphs, and we use the identifiable entities for dataset construction. One orthogonal improvement is to adopt in-domain retrieval corpora for dataset construction in the evidence approach and we leave it for future work.
>
> > In Figure 3, is the color used to represent the gap between the vanilla and FactKB-*? If this is the case, it would be helpful to include this information in the caption to ensure clarity.
>
> Yes, that’s the case. We thank the reviewer for this suggestion and we will definitely include this in the caption.
>
> > In lines 430-431, the citations for the knowledge graphs used are not provided. While they are cited in the appendix, it is also expected that they are cited in the main body of the text.
>
> We will add knowledge graph references in lines 430-431 for the revised version.
>
> We thank the reviewer for their insightful questions. We will add all the clarifications above, especially experimental details about baselines, in the final version of the paper.

---

### Meta-Review · Area_Chair_b8m1 · 2023-09-17

**Recommendation:** 4

**Metareview:**

The paper addresses an important issue of factuality, and propose a novel approach to encode factual knowledge during pre-training. It convincingly shows the merits of their method on both in- and out-of-domain datasets. As suggested by reviewers, I hope that the authors include more qualitative analysis of FactKB in the next version of the paper.

---

### Decision · Program_Chairs · 2023-10-07

**Decision:**

Accept-Main

**Comment:**

The paper addresses an important issue of factuality, and propose a novel approach to encode factual knowledge during pre-training. It convincingly shows the merits of their method on both in- and out-of-domain datasets. As suggested by reviewers, I hope that the authors include more qualitative analysis of FactKB in the next version of the paper.